# Sperm morphology and forward motility are indicators of reproductive success and are not age- or condition-dependent in a captive breeding population of endangered snake

**Mark R. Sandfoss©\*, Steve Reichling, Beth M. Roberts**

Memphis Zoo, Conservation and Research Department, Memphis, TN, United States of America

\* mrsandfo@gmail.com

**Data Availability Statement:** The data underlying the results presented in the study are available

## Abstract

The relationship between male ejaculate traits and reproductive success is an important consideration for captive breeding programs. A recovery plan for the endangered Louisiana pinesnake includes captive breeding for the release of young to the wild. Semen was collected from twenty captive breeding male snakes and ejaculate traits of motility, morphology, and membrane viability were measured for each male. Semen traits were analyzed in relation to the fertilization rate of eggs produced from pairings of each male with a single female (% fertility) to determine the ejaculate factors contributing to reproductive success. In addition, we investigated the age- and condition-dependence of each ejaculate trait. We found significant variation in the ejaculate traits of males and normal sperm morphology ($\bar{x}$ = 44.4 ± 13.6%, n = 19) and forward motility ($\bar{x}$ = 61.0 ± 13.4%, n = 18) were found to be the best predictors of fertility. No ejaculate traits were found to be condition-dependent (*P* > 0.05). Forward progressive movement (FPM) ($\bar{x}$ = 4 ± 0.5, n = 18) was determined to be age-dependent ($r^2$ = 0.27, *P* = 0.028), but FPM was not included in the best model for rate of fertilization. Male Louisiana pinesnakes do not appear to experience a significant decline in reproductive potential with age (*P* > 0.05). The observed average rate of fertilization in the captive breeding colony was below 50% and only those pairings with a male having >51% normal sperm morphology avoided a 0% rate of fertilization. Identification of the factors contributing to the reproductive success of captive breeding Louisiana pinesnakes is of considerable conservation value in the recovery of the species, and captive breeding programs should use assessments of ejaculate traits to plan breeding pairs for maximum reproductive output.

## Introduction

We live in the age of the Anthropocene and biodiversity is decreasing at an alarming rate. *Ex situ* captive breeding programs can be an important part of conservation programs for the preservation and recovery of threatened and endangered species [1, 2]. Captive breeding and

from the corresponding author and are publicly available online at the Figshare repository (https://doi.org/10.6084/m9.figshare.21767276).

**Funding:** This study was funded by the U.S. Fish and Wildlife Service (award #F19AC00617 to SR) and the Conservation Action Network of the Memphis Zoo (awarded to MRS). The funders had no role in study design, data collection and anlaysis, decision to publish, or preparation of the manuscript.

**Competing interests:** The authors have declared that no competing interests exist.

successful development of assisted reproductive technologies, such as artificial insemination, benefit from detailed knowledge of reproductive physiology, and we lack basic information on the factors affecting reproductive success for most wildlife species.

Investigations of reproductive success in natural populations is challenging because of the difficulty in disentangling the many factors that influence success, particularly the role played by male and female factors. In species where males have a higher lifetime reproductive success relative to females, males are predicted to be under intense sexual selection [3]. While sexual selection research has often focused on male traits associated with pre-copulatory selection (e.g., ornamentation and weaponry [4]), there is a growing appreciation for the exaggeration of traits resulting from post-copulatory selection, particularly those associated with male ejaculates [3, 5, 6]. Post-copulatory selection pressure is expected to shape all aspects of male expenditure on ejaculate, including sperm form, function, number, and motility [7–10]. The increased pressure to produce higher quality semen, and associated energetic costs, can translate to condition-dependent sperm quality whereby males in higher body condition produce higher quality sperm [11]. There is an assumption that strong selection would result in uniformly high values among males; however, male fertility in natural populations has been shown to vary considerably among individuals [12]. In the reptile *Anolis sagrei*, males in better body condition have longer sperm and higher sperm counts, changes that also occur in response to the level of competition for mates [13, 14].

Factors affecting reproductive success and post-copulatory selection has received much attention in the contexts of livestock breeding and human infertility treatment. Results of livestock studies attempting to link specific sperm traits to fertility have been contradictory or inconclusive [15–17]. Fertility assessments in humans continue to use analyses of standard semen characteristics including measures of motility, but there is significant evidence that motility is an unreliable predictor of pregnancy [18–20]. Studies of fertilization success in wildlife species have found several aspects of semen characteristics, such as sperm velocity [12, 21], sperm membrane viability [22], and sperm morphology [12], to be indicators of fertility, but no metric of reproductive success has been consistently identified across taxonomic groups. However, our understanding of postcopulatory mechanisms is limited in taxa other than mammals, birds, and insects [3].

In attempting to review squamate sperm competition, researchers [3] found only two comparative analyses of sperm competition in the literature for squamates. In snakes, there is evidence [23] that aspects of sperm morphology and overall sperm size change with increased sperm competition. However, others [24] have found no relationship between the strength of sperm competition and sperm morphology for at least one species of lizard. Therefore, it is difficult to draw strong conclusions for how sperm competition affects sperm characteristics in squamate reptiles. There is a need to investigate reproduction in a wider range of reptile species to truly understand the importance of sperm competition in squamates.

Semen characteristics of snakes have received some attention, but this has primarily been limited to studies focused on cryopreservation (e.g., [25]) and not factors affecting fertilization success. In addition to theoretical considerations of sperm evolution in reptiles, there are practical questions regarding assessment of individuals as breeders in *ex situ* captive breeding programs. An alarming proportion of species of reptile are of conservation concern [26], and we currently know too little about the factors that contribute to reproductive success in reptiles for effective use of *ex situ* captive breeding programs for most reptile species.

An additional factor to consider when attempting to identify drivers of reproductive success in a species is age. Studies in multiple species have found both sperm and seminal fluid to be affected by age and it is generally assumed that reproductive performance of males declines with age [27]. However, there is a need for more comparative studies on additional species

with variable modes of reproduction and life expectancies to improve our understanding of reproductive aging [28–30]. There are striking differences in reproductive aging going from rapid senescence and death after a first breeding season (e.g., salmonids) to negligible reproductive senescence (e.g., naked mole rats, *Heterocephalus glaber*) [28]. Despite recent investigations of reproductive success in lizards [13, 24, 31] and a comparative study of semen quality between two age classes of pit viper [32], male fertility and aging remains understudied in reptiles.

The Louisiana pinesnake, *Pituophis ruthveni*, is a large-bodied, non-venomous colubrid endemic to longleaf pine habitats in Louisiana and Texas and is listed as threatened by the U.S. Fish and Wildlife Service (Federal Register, 2018) and endangered by CITES [33, 34]. A consortium of zoos has established an *ex situ* captive breeding program for the species to produce offspring for reintroduction to the wild. The captive breeding colony provides a unique opportunity to study reproduction in a wildlife species with consideration of individual age and body condition. Males are thought to be sexually mature at 1+ years of age and have been recorded to reproduce in captivity at > 27 years of age. There are no studies of reproduction in the wild for this species despite its imperiled conservation status. There is a published description of breeding season for the Louisiana pinesnake [35], which is based on capture frequency and trap associations of male and female snakes but this does little to advance our understanding of reproductive success. Furthermore, no nest of a Louisiana pinesnake has ever been found in the wild which limits our ability to understand fertility rates of natural clutches and places more emphasis on investigations of captive breeding individuals to characterize important factors for reproductive success. A recent study [36] described the timing and development of follicles in female Louisiana pinesnakes but much remains unknown about male reproductive characteristics. Here, we investigated the relationship between semen characteristics of breeding males and reproductive success in captive Louisiana pinesnakes. We aimed to evaluate the effects of age and body condition on semen characteristics and reproductive success to better understand reproductive ageing in reptiles while developing methods for assessing the reproductive potential of male snakes for the recovery of this species.

## Materials and methods

### Animals and location

Snakes used in this study were 20 pairings of a single male with a single female (*n* = 40 snakes, *n* = 20 clutches) located at two captive breeding facilities near or within their historic range / climate: Fort Worth Zoo (Ft. Worth, Texas, USA) and Ellen Trout Zoo (Lufkin, Texas, USA). Males at Ellen Trout Zoo ranged in age from 4 to 21 years and males at Fort Worth Zoo ranged from 4 to 19 years of age. Photoperiod and temperature were adjusted throughout the annual cycle of breeding animals to mimic natural climate conditions and stimulate reproductive cycles and mating. Snakes were housed individually and fed weekly with commercially produced chicks, rats, or some combination of the two, which was based on size of the snake and feeding behavior of individuals. Water was provided to all snakes *ad libitum*.

To allow for breeding, a male and a female Louisiana pinesnakes were placed together in a single large cage for at least 30 days starting in March (Fort Worth Zoo) or April (Ellen Trout Zoo). Snakes were separated in June and July, and females were provided with a nest box for oviposition. Differences in dates of introduction and separation were based on temperature cycle at each breeding location. Females included in the study had incomplete breeding histories.

The research project was conducted as part of the Association of Zoos and Aquarium's Species Survival Plan for the Louisiana pinesnake, and all research was approved by an Institutional Animal Care and Use Committee (IACUC #2020–4, IACUC #2021-01-19).

## Incubation of eggs

Once eggs were laid, they were removed from their nest box and gently placed in an incubation box. Incubation boxes were 12" x 12" x 4" plastic containers filled with a 1:1 ratio of vermiculite to water. Each incubation box contained a single clutch from a female, and each egg was candled at least once during the incubation period to confirm fertilization. Incubation boxes were housed inside larger incubators to maintain temperature and humidity. Eggs were incubated at a temperature of 25–30°C until hatching which occurs ~66 days from oviposition. Fertilization of eggs was determined via the visible presence of an embryo in an egg which presented as a dark spot with distinct venation on the inside of the egg during candling. Emergence of a neonate snake at the end of the incubation confirmed fertilization. Percent fertility was calculated as the percentage of eggs laid that were successfully fertilized [37] and was used in statistical tests to identify male semen characteristics that contributed significantly to reproductive success.

Twenty females that were each paired with a single male laid a clutch of eggs that ranged between 2 and 10 eggs ($\bar{x}$ = 6.4 ± 2.4). Fertilization rates ranged between 0 and 100% with the average rate being 47.9 ± 41% (Table 1). No females double clutched in 2021 and males were only paired with a single female.

## Semen collection methods

Semen was collected for assessment using ventral massage [38–40] from seven males (mass range 920–1426 g) from seven breeding pairs at Ellen Trout Zoo and thirteen males (mass range 945–3180 g) from thirteen pairings at Fort Worth Zoo. Briefly, semen collection involved placing individuals in a bucket with a small amount of water at room temperature prior to semen collection to encourage defecation. Manual expression of semen was achieved by applying pressure to the ventral side of the snake starting at mid-body and slowly moving towards the cloaca. Semen was expelled from the vas deferens into the open cloaca without the need to evert the hemipenes. Semen was collected from the cloaca with a micropipette and transferred to a tube containing 50 μl of "H10" which consisted of TL Hepes (Caisson Laboratories Inc. #IVL01) supplemented with 10% Fetal Bovine Serum (v/v, Sigma-Aldrich #F7524). Typically, 1–5 fractions of ejaculate were collected from each snake per session. Fractions were stored on ice at 4°C from time of collection until processing ($\leq$ 2 h).

## Semen analyses

Fractions were initially assessed for volume, color (tan, yellow, white, clear), and texture (thin, thick, flaky) during collection and those fractions found to be clear of contaminants and containing motile sperm were combined for each individual. The combined semen sample was

**Table 1. Summary table of male traits and fertilization rate of pairings from assessments of 20 captive breeding Louisiana pinesnakes.**

| Parameter | Ellen Trout Zoo | | | Fort Worth Zoo | | | Combined | | |
|---|---|---|---|---|---|---|---|---|---|
| | Mean | SD | N | Mean | SD | N | Mean | SD | N |
| Mass (g) | 1232 | 184.3 | 7 | 1826.5 | 664 | 13 | 1618.5 | 611.4 | 20 |
| Age (years) | 13.7 | 7.6 | 7 | 8 | 5.3 | 13 | 10 | 6.6 | 20 |
| % Forward Motility | 62.2 | 10.9 | 6 | 60.4 | 14.9 | 12 | 61 | 13.4 | 18 |
| % Total Motility | 71.7 | 9.1 | 6 | 69.4 | 12.4 | 12 | 70.2 | 11.2 | 18 |
| FPM Index | 3.9 | 0.6 | 6 | 4 | 0.5 | 12 | 4 | 0.5 | 18 |
| Concentration x $10^6$ per ml | 986.7 | 712.2 | 5 | 619.6 | 517.7 | 13 | 721.6 | 580.6 | 18 |
| % Membrane Viability | 84.7 | 10.3 | 6 | 66.4 | 18 | 13 | 72.2 | 17.9 | 19 |
| % Normal Morphology | 54.7 | 14.5 | 6 | 39.6 | 10.6 | 13 | 44.4 | 13.6 | 19 |
| % Acrosome Integrity | 58.7 | 14.2 | 6 | 41.2 | 11 | 13 | 46.7 | 14.3 | 19 |
| % Fertility | 57.6 | 42.6 | 7 | 42.7 | 40.6 | 13 | 47.9 | 40.8 | 20 |

then diluted to a total of 1:8 to 1:10 (semen: total volume) with H10 depending on combined sample volume and sperm concentration.

Fresh diluted samples were then immediately assessed for several metrics of semen characteristics. Sperm motility was characterized by the number of individual sperm that were moving/ moving forward/ non-moving and percent total motility and percent forward motility calculated for 100 randomly observed sperm. Simultaneously, FPM was assigned on a scale of 0 to 5 (scale 0 = non-motile, 5 = fast and straight motility). Sperm concentration of fixed sperm (1% paraformaldehyde / saline) was determined using a hemocytometer (Bright-Line, American Optical Corporation). Percent membrane viability of sperm was determined using an eosin-nigrosin-based live-dead stain (Jorvet Stain, Jorgensen Laboratories, Inc., Loveland, CO, USA) and counting the first 100 sperm. Sperm morphology and acrosome integrity was assessed by Pope's stain [41]. Morphology of the first 200 sperm encountered was categorized as having "normal" or "abnormal" morphology by examining the acrosome, sperm head, midpiece, and tail for abnormalities including swollen head, bent midpiece, broken tails, membrane droplets, and additional abnormalities (Fig 1). Sperm without abnormalities were considered normal. Acrosome integrity of 200 individual spermatozoa was categorized as absent, present, or damaged. All assessments of spermatozoa were completed using either an Olympus BX60 or CX41 phase contrast microscope (motility, FPM, and plasma membrane integrity at a magnification of 400x, concentration at 400x, and morphology and acrosome integrity at 1000x under oil immersion).

## Body condition index

We measured snout-vent length (SVL, cm) and body mass (g) for all snakes included in the study. We measured SVL using analyses of pictures with the software ImageJ. An index of body condition (BCI) was calculated for each male as the residual of the linear regression of log-body mass (g) and log-SVL (cm) for the 20 male snakes included in the study.

## Statistical analyses

Percent fertility was used in statistical models as our measure of reproductive success for males. We investigated the effects of 10 male characteristics on fertilization success (% fertility): age, % forward moving sperm, % total sperm motility, FPM, % sperm viability, % sperm with normal morphology, % sperm with acrosome integrity, concentration of sperm, male BCI, and breeding location. We tested for an effect of each male characteristic on % fertility using linear regression models by adding independent variables and looking for significant terms. We added and removed independent variables and combinations of up to three variables at a time to investigate effects of each and then used AICc model selection to select the final model with the lowest AICc value.

Pearson correlation tests were used to test for linear correlations between age and each of seven male semen characteristics measured (% forward motility, % total motility, FPM Index, sperm concentration, % viability, % normal morphology, and % acrosome integrity) and BCI with each of the same seven male characteristics. We also tested for correlation between age and BCI using a Pearson correlation test. We used a Student's T-test to detect any differences in BCI between locations. All statistical tests were performed using program R (version 3.6.0) and alpha was set at 0.05.

## Results

### Semen collections

Semen was successfully collected from all breeding males; although not all individuals (n = 3) produced samples of sufficient quantity to assess all semen characteristics. There was considerable variation observed in semen characteristics of males (Table 1).

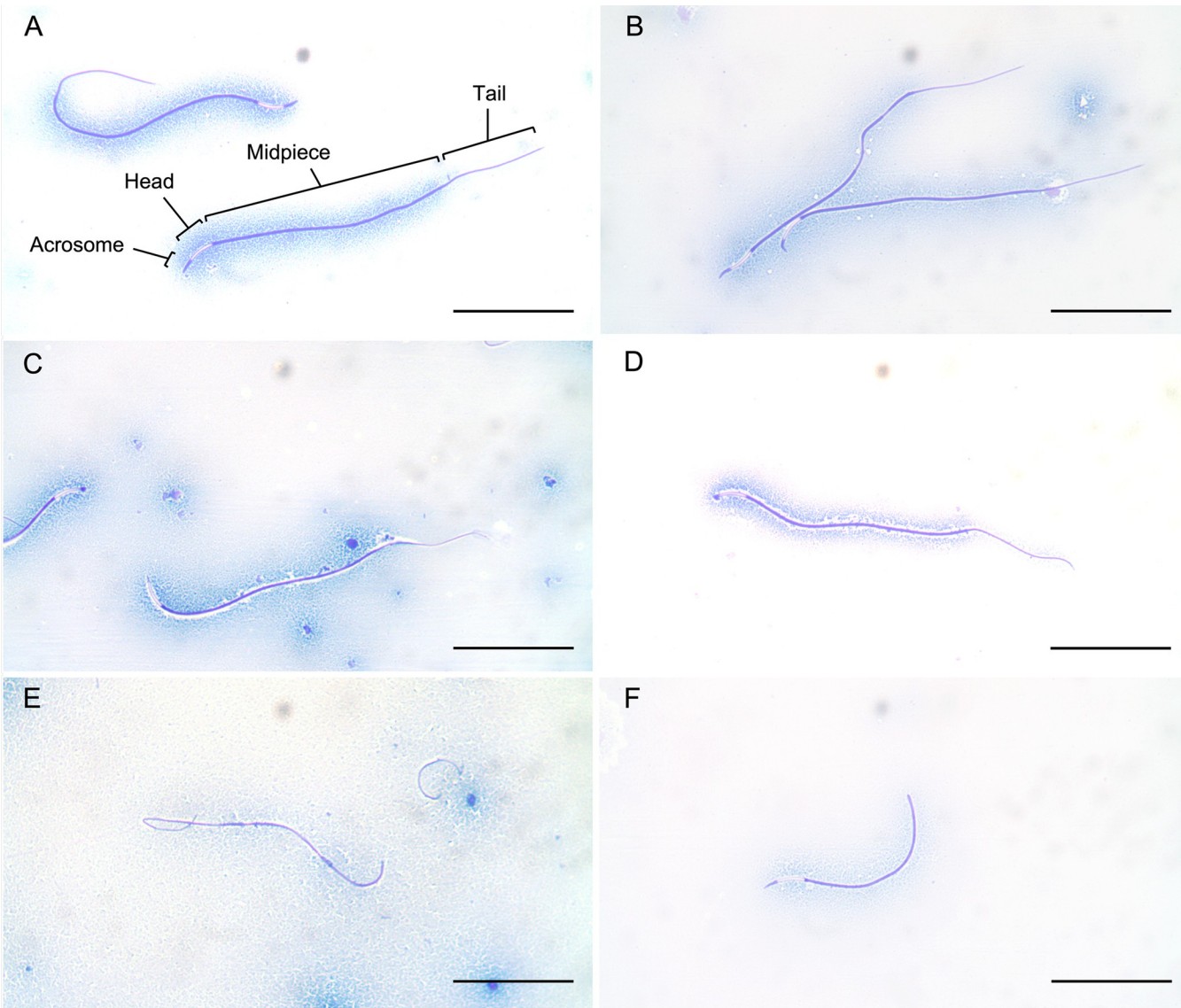

**Fig 1.** Microscopic images of sperm from the Louisiana pinesnake stained using a Pope's stain on sperm with A) normal morphology and sperm demonstrating abnormalities including B) bent acrosome, C) missing acrosome, D) shortened acrosome, E) missing head, and F) missing tail. All images taken at 1000X magnification using an Olympus 0X40 phase-contrast microscope. Bars = 50 μm.

### Reproductive success and male characteristics

The final model with the lowest AICc value identified normal sperm morphology as having a significant positive effect on fertility rate ($F_{1,14} = 6.04$, $P = 0.028$) (Fig 2A) while forward motility had a marginally positive effect on fertilization ($F_{1,14} = 2.68$, $P = 0.124$) (Fig 2B). The location of breeding snakes was not included in any of the top models (Table 2) and no significant interactive effects were detected.

We observed considerable variation in BCI values for males and no difference (Student's t-test, t = –0.79, d.f. = 18, $P = 0.44$) was found between the BCI of males from Fort Worth Zoo ($\bar{x} = 0.01 \pm 0.09$) and Ellen Trout Zoo ($\bar{x} = –0.02 \pm 0.05$). Correlation tests did not identify any instances of condition dependence as no significant relationships were found between BCI and semen characteristic measured (Fig 3). Age was not found to be a significant predictor of

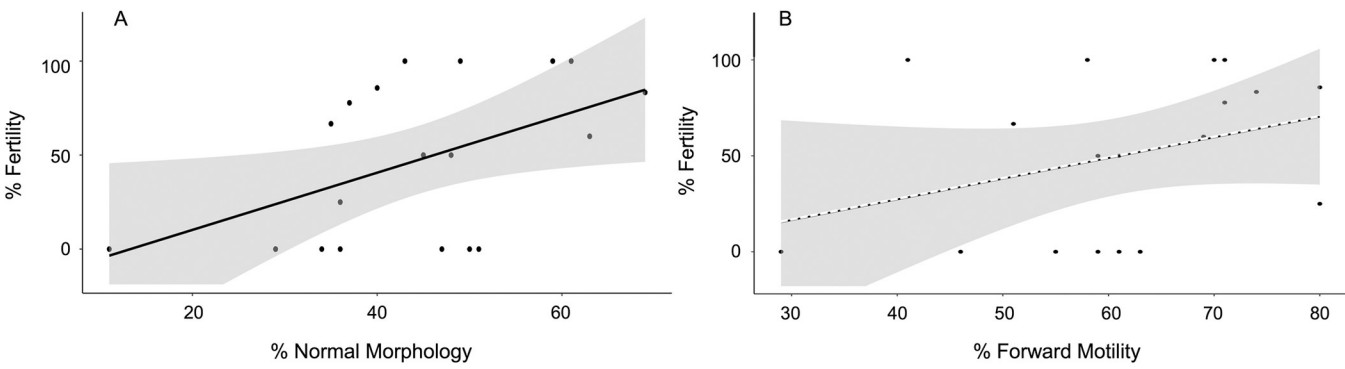

**Fig 2.** Scatterplots of the linear relationship between A) percent sperm with normal morphology and percent fertility ($r^2 = 0.24$, $P = 0.03$) and B) percent sperm with forward motility ($r^2 = 0.12$, $P = 0.16$) of semen from 20 adult male Louisiana pinesnakes. Solid line represents a significant regression line at the 0.05 level, while a dashed line represents a non-significant regression line and shading shows the 95% CI.

male fertility in the model of best fit, but increasing age significantly correlated with reduced FPM ($r^2 = 0.27$, $P = 0.028$) and marginally with decreasing concentration of sperm ($r^2 = 0.14$, $P = 0.131$) (Fig 4).

## Discussion

Individual semen characteristics of males have a significant effect on rate of fertilization in captive breeding Louisiana pinesnakes. There was considerable variation in the ejaculate quality of male snakes. Males that produced semen of higher quality, as characterized by a greater percentage of sperm with normal morphology and forward motility, had higher levels of reproductive success. Interestingly, sperm morphology has been linked to reproductive success in some species such as humans and domestic livestock [42–44], but for others [8, 45, 46]. Such conflicting results across taxonomic groups raises several interesting questions regarding which traits selection is acting on during post-copulatory selection and what other factors influence ejaculate traits (e.g., reproductive mode—summarized for snakes by [23]. Recent investigations of mammalian reproduction have found that the movement of sperm through the female reproductive tract is regulated by a complex array of female tract factors which may be the mechanism for selection on sperm of normal morphology and optimal motility to reach the egg [47]. There remains much we do not understand about the evolution of sperm form and function and the role of sexual selection, particularly in reptiles [47, 48].

The Louisiana pinesnake is fossorial and females have relatively small home ranges, but males are known to make large distance movements [49, 50]. Males have been found in traps with females indicating males have some ability to find mates [35]. It is probable that females will mate with multiple males in the wild which would presumably lead to sperm competition

**Table 2. Summary table of top Anova models based on lowest AICc scores for fertilization rate in captive breeding Louisiana pinesnakes (n = 20).**

| Model | AICc | Model Parameters | F value | DF | *P* value |
|---|---|---|---|---|---|
| Model 1 | 21.32 | % Forward Motility | 2.68 | 1, 14 | 0.124 |
| | | % Normal Morphology | 6.04 | 1, 14 | 0.028* |
| Model 2 | 22.07 | % Normal Morphology | 5.49 | 1,17 | 0.032* |
| Model 3 | 22.73 | % Forward Motility | 2.46 | 1,14 | 0.139 |
| | | % Normal Acrosome | 4.43 | 1,14 | 0.054 |

*statistically significant at alpha level of 0.05.

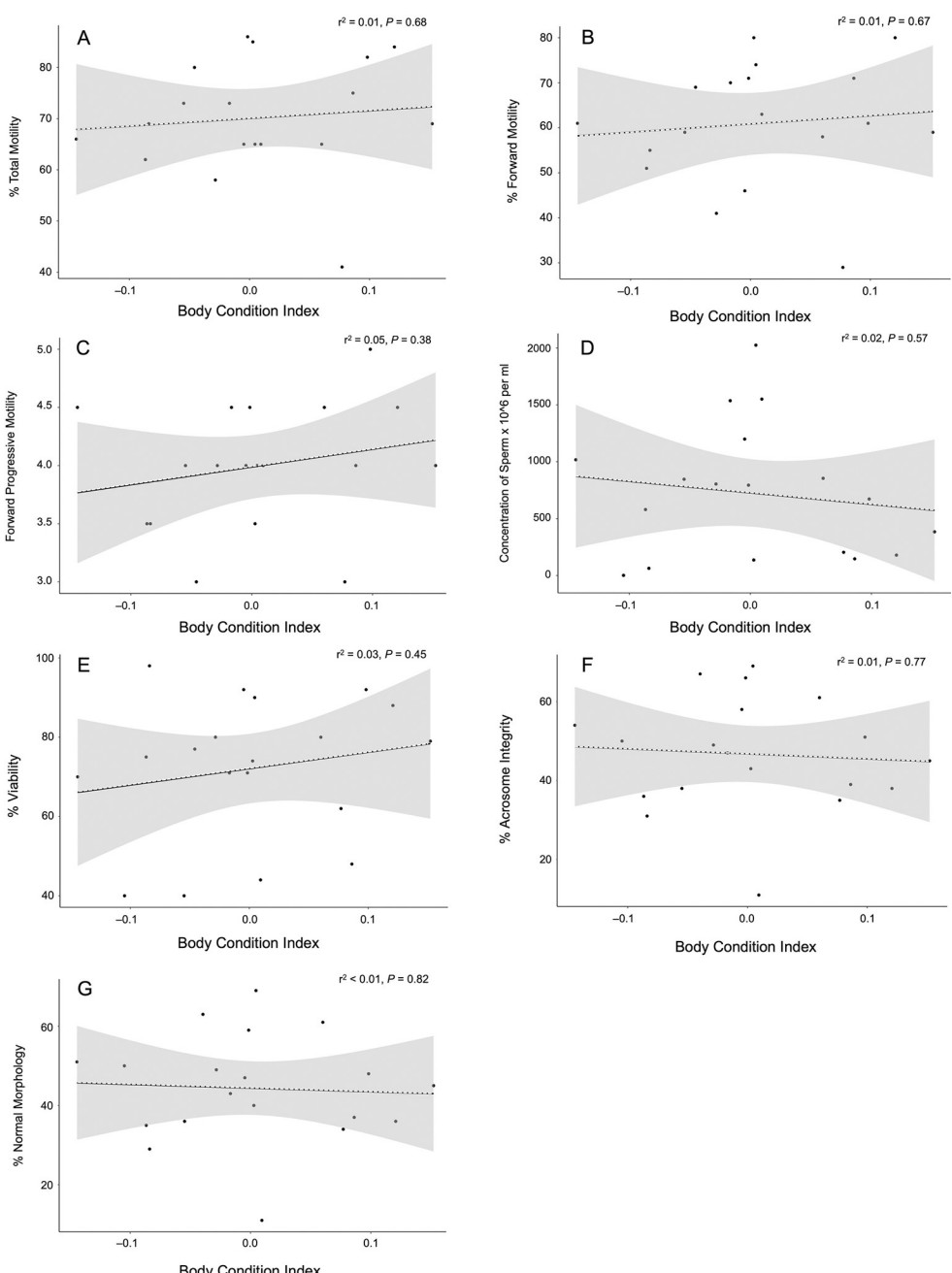

**Fig 3.** Scatterplots of the linear relationship between A) body condition index (BCI) and percent total motility, B) BCI and percent forward motility, C) BCI and forward progressive motility, D) BCI and concentration of sperm, E) BCI and percent viability of sperm, F) BCI and percent acrosome integrity, and G) BCI and percent normal morphology of semen from 20 adult male Louisiana pinesnakes. Dashed lines represent a non-significant regression line at the 0.05 level and shading shows the 95% CI.

[9, 23, 51]. Our results provide support for post-copulatory selection on semen quality in reptiles.

The role of cryptic female choice following copulation [52] is unknown in the Louisiana pinesnake. Sperm storage does occur in snakes, and we cannot discount the possibility of its occurrence here but, based on our experience with captive breeding and artificial insemination

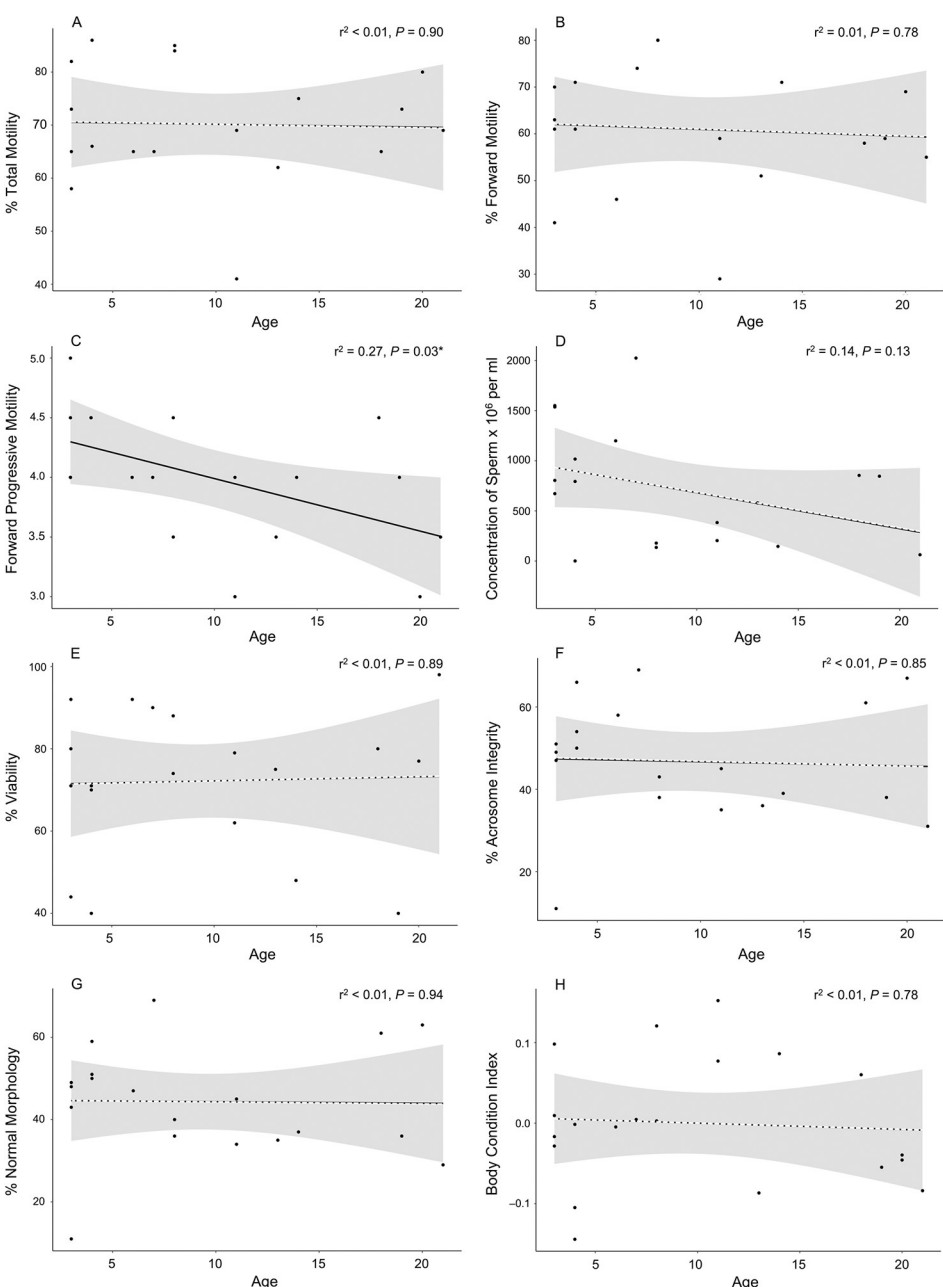

**Fig 4.** Scatterplots of the linear relationship between A) age and percent total motility, B) age and percent forward motility, C) age and forward progressive motility, D) age and concentration of sperm, E) age and percent viable sperm, F) age and percent normal acrosome, G) age and percent normal morphology of semen, and H) age and body condition of 20 adult male Louisiana pinesnakes. Dashed lines represent a non-significant regression line at the 0.05 level and shading shows the 95% CI.

experiments (MRS Unpublished Data), sperm storage is rare in this species. For example, in each of seven years of records from three Louisiana pinesnake breeding facilities, females that were not paired with a male during a breeding season laid unfertilized eggs that year, even if mated in previous seasons, whereas there has been no instance of a non-bred female laying fertilized eggs.

While it can be assumed that females contribute to the fertilization rates of eggs, it is beyond the scope of this study to measure all possible factors that contribute to reproductive success, and we instead focus here on ejaculate traits of males. Captive breeding males showed a considerable range in values of normal sperm morphology (11% to 69%). Normal sperm morphology is a somewhat generic phrase to describe the physical aspect of all parts of a sperm (reviewed in [53]), and our assessments included examination of the acrosome, head, midpiece, and tail (Fig 1). Successful fertilization is assumed to require normal morphology of each of these components. Measuring the effects of variation in sperm morphology on sperm performance is complex [54], but morphology is known to affect motility of sperm, such as velocity [31, 55, 56]. Normal acrosome function is necessary for the head to penetrate the zona pellucida of an ovum and deliver its contents—the haploid genome of the sire and a nongenetic payload of RNAs and proteins [57]. Despite the significant finding for an effect of sperm morphology on fertilization rate, we found considerable overlap in the values of normal sperm morphology across males whose pairings produced both fertile and infertile eggs making the identification of a threshold for infertility difficult. For example, one male with a relatively low percentage of sperm with normal morphology (35%) successfully fertilized multiple eggs (67% fertility). We can, however, set a threshold for fertility as all male snakes with normal morphology of >51% fertilized at least one egg laid by their mate (>0% fertility).

The primary deformity of sperm we observed in males was a deformed or missing acrosome, but two males had sperm with detached heads as their main deformity. Sperm morphology can change seasonally (e.g., [58]), but the mechanisms behind poor sperm morphology are generally understudied in all taxonomic groups, apart from laboratory mice and humans (e.g., [59], see review by [60]). As such, it is not currently possible to determine the causes of acrosomal abnormality and infertility [61] and limits our ability to mediate sperm deformity in our captive breeding population.

The captive breeding population of Louisiana pinesnakes was started with 19 founder individuals from both wild and captive sources. Inbreeding is a concern with captive breeding programs and small populations with varying degrees of fertility can be a result of inbreeding [62–64]. In this study, seven mating pairs (out of 20) produced zero fertile eggs. We are not able to determine all the factors that might have contributed to the low level of reproductive success in captive animals. In humans, where sperm characteristics have been the most intensively studied, fertility issues can be caused by multiple factors including genetic abnormalities, mitochondrial mutations, sexual abstinence, lifestyle and environment, radiation, heat exposure, pollution, stress, infections, among others (reviewed in [65]). Unfortunately, the Louisiana pinesnake is secretive and seldom observed in nature. There is little known about the reproductive activity of this species in the wild and no oviposition site has ever been observed for the Louisiana pinesnake.

The few field studies that have addressed infertility in wild populations have shown that reduced male fertility or temporary male infertility may be more common than previously thought [66, 67], but these studies have been limited to mammals and birds. There are no known studies of fertility in wild snakes. In one study of wild lizards (*Lacerta agilis*) [37], found little evidence of infertility in males and observed a relatively high fertilization rate of 76.7 ± 29.4%. The low fertility rate of captive breeding Louisiana pinesnakes in our study (47.9 ± 40.8%) is concerning, but more information on fertility and fertilization rates in wild reptile populations, particularly snakes, is needed before any meaningful comparisons can be drawn.

Infertility of males can result from environmental stressors such as food scarcity or pathogens (reviewed in [68]). While food availability is not an issue in our breeding colony of snakes, captivity itself may present a significant stressor to snakes. We did see considerable

variation in the BCI values of males. Body condition is often used as a measure of health and fitness in wild animals, and individuals in better body condition are assumed to be healthier and have higher lifetime reproductive output. We predicted that the potential selection pressure on semen quality of males would lead to condition-dependent semen characteristics which would be directly linked to reproductive success [13]. However, we found no evidence that males with higher BCI produced higher quality semen. Investigations of body mass and reproduction have not always found male body mass to be an indicator of reproductive success (e.g., roosters, *Gallus gallus*, [69]). Clearly, the connection between individual body condition and ejaculate quality requires further investigation in reptiles as does the mechanisms driving variation in ejaculate traits of Louisiana pinesnakes.

Age was not found to be a significant predictor of male fertility in our study. Interestingly, we found increasing age to significantly correlate with reduced FPM and marginally with concentration of sperm in semen which agrees with previous study [70], but neither of these metrics of sperm quality were identified as important predictors of reproductive output in Louisiana pinesnakes. The declines we observed in FPM do not appear to be a strong factor in reproductive output based on fertilization success. This result agrees with other studies that have found no effect of age on reproductive output in a captive breeding setting (e.g., whooping crane, *Grus americana*, [71]), while age was found to be a factor in captive breeding of the red wolf, *Canis rufus* [72]. Male pinesnakes reproduced in captivity up to ~28 years old (SR Unpublished Data). The influence of male age on aspects of semen quality is inconsistent across taxonomic groups whereby in some species younger males have poorer quality semen (cheetahs, *Acinonyx jubatus*, [73]; Baird's tapir, *Tapirus bairdii*, [74]) while in others younger males have better quality semen (rats, *Rattus norvegicus*, [75]; humans, *Homo sapiens*, [76]). To further complicate matters, a study of reproductive fitness in lizards found males of an intermediate age produced offspring of the highest viability [77]. The only other study of semen quality and age in snakes [32] found seasonal and age effects on measures of sperm motility and concentration in the pit viper *Bothrops insularis*, but how these factors affected rates of fertilization was not measured. It should be noted that reproductive ageing does not affect both sexes equally [78–81], and female age, which we did not consider here, might contribute to reproductive success in Louisiana pinesnakes. Overall, there is considerable variation across species [82], and further studies are required in both captive and wild reptiles to better understand reproductive aging.

The recovery of the Louisiana pinesnake requires a multi-faceted approach that includes a successful captive breeding program. The identification of sperm morphology as a predictor of male reproductive output in this species will provide significant value to captive breeding efforts, and sperm morphology should be assessed in all males involved in captive breeding programs for other snake species of conservation concern. Further study across a wider taxonomic breadth of reptiles is needed to improve our understanding of the male and female factors that affect reproductive success in captivity and the wild. This is particularly important in the context of conservation as a considerable number of reptiles are threatened with extinction [26].

## Acknowledgments

We would like to acknowledge the assistance of staff at the Ellen Trout Zoo including Gordon Henley, Charlotte Henley, Robert Jackson, Mike Nance and at the Fort Worth Zoo particularly Diane Barber, Vicky Poole, Jaimie Galm, and Valeria Gladkaya. We would also like to thank the efforts of additional staff and volunteers at the Memphis Zoo including Jessica Cantrell, Melanie Richter, Gary Wigman, Palmer Mihalevich, Crista Fennessee, and Rita DeLucco. We

would also like to acknowledge comments provided by Alexandre Rodrigues Silva that greatly improved the manuscript.

## Author Contributions

**Conceptualization:** Mark R. Sandfoss.

**Data curation:** Mark R. Sandfoss.

**Formal analysis:** Mark R. Sandfoss.

**Funding acquisition:** Mark R. Sandfoss, Steve Reichling.

**Investigation:** Mark R. Sandfoss, Beth M. Roberts.

**Methodology:** Mark R. Sandfoss, Beth M. Roberts.

**Project administration:** Mark R. Sandfoss, Steve Reichling.

**Resources:** Beth M. Roberts.

**Supervision:** Steve Reichling.

**Validation:** Mark R. Sandfoss.

**Writing – original draft:** Mark R. Sandfoss.

**Writing – review & editing:** Steve Reichling, Beth M. Roberts.

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
