## [Decision Letter · Decision Letter 0]

22 Nov 2022

PONE-D-22-29909Sperm morphology and forward motility are indicators of reproductive success and are not age- or condition-dependent in a captive breeding population of endangered snake.PLOS ONE

Dear Dr. Sandfoss,

Thank you for submitting your manuscript to PLOS ONE. After careful consideration, we feel that it has merit but does not fully meet PLOS ONE’s publication criteria as it currently stands. Therefore, we invite you to submit a revised version of the manuscript that addresses the points raised during the review process. Article must be revised by authors and add a better figure 1. This type of reproductive tool and specie must be highlighted during introduction.

We look forward to receiving your revised manuscript.

Kind regards,

Carlos E. Ambrósio, Ph.D

Academic Editor

PLOS ONE

Journal Requirements:

"Funding acquisition: SR and MRS"

"This study was funded by the U.S. Fish and Wildlife Service (award #F19AC00617 to SR) and the Conservation Action Network of the Memphis Zoo (awarded to MRS). 

The funders had no role in study design, data collection and anlaysis, decision to publish, or preparation of the manuscript."

Reviewers' comments:

Reviewer's Responses to Questions

**Comments to the Author**

1. Is the manuscript technically sound, and do the data support the conclusions?

Reviewer #1: Yes

2. Has the statistical analysis been performed appropriately and rigorously? 

Reviewer #1: Yes

3. Have the authors made all data underlying the findings in their manuscript fully available?

Reviewer #1: Yes

4. Is the manuscript presented in an intelligible fashion and written in standard English?

Reviewer #1: No

5. Review Comments to the Author

Reviewer #1: This is an interesting study related to the reproductive performance of a captive breeding population of Louisiana pinesnake. This kind of studies are extremely important for the establishment of strategies focused on the conservation of wild species like that. Manuscript, however, lacks some major revisions.

Tittle - adequate

Abstract - This is well written in general. I miss some numeric results related to mean values for progressive motion and correlations.

Keywords - ok

Introduction - Even if introduction brings an excellent background related to male importance for the reproductive fitness of a species in general, these concepts were mainly based in studies on mammals, specially those related to livestock. Authors are advised to provide some background related to the knowledge on reptile reproduction. Moreover, please provide some background related to the pinesnake reproduction. what is known about the reproduction of this species so far? what bottleneck is intended to be filled with this study?

Methodology:

- At what age do animals reach puberty? At what age are they considered adults? At what age are they considered senescent? This information might be useful to divide the animals into groups and also to understand that the experimental group would be formed by individuals in different reproductive stages. In this regard, we would need to understand how many animals of each reproductive stage were used in the composition of the experimental group.

- In the beginning of the section, authors indicated that they used 40 individuals, but they only report the semen collection from 20 snakes (seven from Ellen Trout Zoo and 13 from Fort Worth Zoo). Moreover, the weight from the animals seems to vary a lot, showing that they would probably not be a homogeneous group, maybe due to different ages. Please, clarify these subjects.

- Please provide references for the semen collection technique and for all the methods used in the study.

- in statistical analysis, it is not clear how Fertility rate was evaluated since authors did not provided any methodology for this kind of evaluation before.

- Please specify the variables used for the the regression or correlation analysis. This is not clear.

Results

- At the end of the table 1, authors present the Fertility parameter. What does it means? Please explain in methodology how this parameter was obtained.

- Eggs produced: authors should clarify this at the methodology. I believe that the detainment of this methodology is very important to be presented at the methods section. In my point of view, authors should clarify the origin of the females, since some snakes have the ability to store sperm from previous copulations, thus interfering in the fertility rates.

- Please provide better resolution images for Figure 1.

Discussions and conclusions are adequate

6. PLOS authors have the option to publish the peer review history of their article (what does this mean?). If published, this will include your full peer review and any attached files.

Reviewer #1: **Yes: **Alexandre Rodrigues Silva

---

## [Author Response · Author response to Decision Letter 0]

6 Jan 2023

A separate Word document has been uploaded with our responses to the comments provided by the Reviewer. We can also paste that information here but formatting may be compromised.

Reviewers' comments:

Reviewer's Responses to Questions

Comments to the Author

1. Is the manuscript technically sound, and do the data support the conclusions?

Reviewer #1: Yes

2. Has the statistical analysis been performed appropriately and rigorously? 

Reviewer #1: Yes

3. Have the authors made all data underlying the findings in their manuscript fully available?

Reviewer #1: Yes

4. Is the manuscript presented in an intelligible fashion and written in standard English?

Reviewer #1: No

5. Review Comments to the Author

Reviewer #1: This is an interesting study related to the reproductive performance of a captive breeding population of Louisiana pinesnake. This kind of studies are extremely important for the establishment of strategies focused on the conservation of wild species like that. Manuscript, however, lacks some major revisions.

Thank you for the review of our manuscript. We appreciate your time and effort to improve our work.

Tittle – adequate

Thank you.

Abstract - This is well written in general. I miss some numeric results related to mean values for progressive motion and correlations.

We appreciate your suggestion and we have added means for several measures and statistics for significant findings to the abstract. We have also edited the text to improve overall clarity.

Keywords - ok

Thank you.

Introduction - Even if introduction brings an excellent background related to male importance for the reproductive fitness of a species in general, these concepts were mainly based in studies on mammals, specially those related to livestock. Authors are advised to provide some background related to the knowledge on reptile reproduction. 

We appreciate the suggestion of the reviewer. Our use of studies on mammals and livestock is based primarily on the availability of studies in the literature. Our understanding of postcopulatory mechanisms of selection is limited in taxa other than mammals, birds, and insects. While there is information on basic aspects of reproduction for a few species of reptiles, we are missing a considerable amount of detail for the majority of species. We have added text to the introduction to provide a better summary of our current understanding of sperm selection and presented the gaps in knowledge that exist for semen analyses of male snakes.

Moreover, please provide some background related to the pinesnake reproduction. 

There is limited published information on the reproduction of Louisiana pinesnakes. However, we have edited the introduction to include all known information including a recent study of follicle development (Oliveri et al., 2022) which was published since the original submission of this manuscript.

what is known about the reproduction of this species so far? 

Very little. We have better highlighted the status of our knowledge in the introduction based on your suggestion.

what bottleneck is intended to be filled with this study?

We believe that this study makes a meaningful contribution to the scientific literature of reproduction in reptiles in two areas: 1) an investigation into the drivers of fertility in male snakes to test evolutionary hypotheses related to the effects of age and body condition and 2) answer practical questions related to assessing the quality of male snakes for breeding in an ex situ recovery program. This information has been added to the introduction and appreciate your suggestion that we provide missing justification for the value of our study.

Methodology:

At what age do animals reach puberty? At what age are they considered adults? At what age are they considered senescent? 

We believe these to be interesting questions that are somewhat difficult to answer for snakes but will consider here. Male Louisiana pinesnakes have shown the presence of sperm in semen collections at 12 months of age but it is difficult to determine when a male is sexually mature. Males may be producing sperm but considering their small stature relative to females, which tend to be larger, they are probably lacking the physical ability to subdue and inseminate a “sexually mature” female of much larger size. However, a single 1 year old male was documented to have produced a clutch of viable offspring within a zoo setting. Females have been observed to reproduce in captivity at 2 years of age, but 3 years is much more typical and often what we consider to be a mature animal. 

Senescence is very hard to discern in snakes. Based on our experience with captive breeding snakes, we often find that snakes will experience mortality from illness or succumb to complications related to reproduction, rather than die of “old age” after reaching a non-reproductive state. We are not aware of any in-depth study on ageing or reproductive senescence in wild reptiles, particularly snakes. As noted in Hoekstra et al. (2020), our understanding of senescence in reptiles is still in its infancy.

Summary of age at reproduction - Louisiana pinesnake captive breeding program (1984 to 2022)

Sire: Min = 0.82 years, Max = 27.93, Mode = 10.92

Dam: Min = 1.92 years, Max = 21.09, Mode = 7.03

Hoekstra, L. A., Schwartz, T. S., Sparkman, A. M., Miller, D. A. W., & Bronikowski, A. M. (2020). The untapped potential of reptile bio-diversity for understanding how and why animals age. Functional Ecology, 34, 38-54.

This information might be useful to divide the animals into groups and also to understand that the experimental group would be formed by individuals in different reproductive stages. In this regard, we would need to understand how many animals of each reproductive stage were used in the composition of the experimental group.

All animals in the study were considered adults. We hesitate to group individuals into reproductive categories for the simple fact that we know too little about reproductive stage in adult male snakes in particular reference to senescence. We are certain that none of our males were immature. One of the goals of this study was to identify the effects of ageing on semen quality of males. There appears to be little to no effects of age based on our results and would lend further support to not grouping males by a reproductive stage. We understand the interest of the reviewer, but we believe the study is better served to use male age as a continuous variable rather than a category of reproductive stage.

- In the beginning of the section, authors indicated that they used 40 individuals, but they only report the semen collection from 20 snakes (seven from Ellen Trout Zoo and 13 from Fort Worth Zoo). 

We apologize for any confusion we have caused. Because females were part of the study in terms of using their eggs, we had a total of 40 individuals consisting of 20 pairings of one male and one female. Therefore, semen was collected from the 20 males. We have edited the text for clarity and Line 155 of revised manuscript now reads “Snakes used in this study were 20 pairings of a single male and female snake (n = 40 snakes, n = 20 clutches) located at two captive…”.

Moreover, the weight from the animals seems to vary a lot, showing that they would probably not be a homogeneous group, maybe due to different ages. Please, clarify these subjects.

We appreciate the interest of the reviewer regarding the variable weights of male snakes. There is certainly a wide range of male weights included in this study. We believe this is a positive aspect of the study as it allowed us to robustly evaluate the role, if any, of body condition on semen characteristics and ultimately fertilization success. One of our correlation analyses included a test of the correlation between BCI (related to weight) and age and found no strong relationship (Fig. 4H).

- Please provide references for the semen collection technique and for all the methods used in the study.

It was Mengden et al. (1980) that originally developed the semen collection technique that was later modified and used by Fahrig et al. (2007) in corn snakes and Sandfoss et al. (2021) in Louisiana pinesnakes. These references have been cited in the methods section where we describe semen collection.

- in statistical analysis, it is not clear how Fertility rate was evaluated since authors did not provided any methodology for this kind of evaluation before.

We apologize for the oversight. We have tried to make it clear how we calculated fertility rate and we have moved up our description of fertility rate earlier in the manuscript. Thank you for bringing this to our attention. 

Line 183 (revised) of Methods section now reads: Percent fertility was calculated as the percentage of eggs laid that were successfully fertilized (Olsson and Shine, 1997) and was used in statistical tests to identify male semen characteristics that contribute significantly to reproductive success.

- Please specify the variables used for the the regression or correlation analysis. This is not clear.

We understand why the reviewer was confused by our language in the previously written version of the statistics section. We have edited the text to make it clear what tests were conducted on which variables. We have also listed that information here for the reviewer.

Correlation tests of BCI with: % total motility, % forward motility, FPM, sperm concentration, % viability, % acrosome integrity, % normal morphology.

Correlation tests of age with: % total motility, % forward motility, FPM, sperm concentration, % viability, % acrosome integrity, % normal morphology.

Correlation test of age with BCI.

We used linear regression models to test for effects of 10 male characteristics on the response variable (% fertility). The 10 male characteristics we tested for inclusion in models were: age, % forward moving sperm, % total sperm motility, FPM, % sperm viability, % sperm with normal morphology, % sperm with acrosome integrity, concentration of sperm, male BCI, and breeding location.

Results

- At the end of the table 1, authors present the Fertility parameter. What does it means? Please explain in methodology how this parameter was obtained.

We have moved this information to the methods section concerning “Incubation of eggs”. The text now reads, “Fertilization of eggs was determined via the visible presence of an embryo in an egg which presented as a dark spot with distinct venation on the inside of the egg during candling. Emergence of a neonate snake at the end of the incubation confirmed fertilization. Percent fertility was calculated as the percentage of eggs laid that were successfully fertilized (Olsson and Shine, 1997) and was used in statistical tests to identify male semen characteristics that contributed significantly to reproductive success.” 

- Eggs produced: authors should clarify this at the methodology. I believe that the detainment of this methodology is very important to be presented at the methods section. 

Thank you for the suggestion. We have moved this section of text up to the methods section under “Incubation of eggs”.

In my point of view, authors should clarify the origin of the females, since some snakes have the ability to store sperm from previous copulations, thus interfering in the fertility rates.

We understand the reviewer’s concerns regarding the origin of females. Unfortunately, we cannot account for the mating history of females. The captive breeding population originates from only 19 founders that were wild-caught or purchased from private collectors. There are simply not enough virgin females located at zoos to conduct a study that would not include previously mated females. We have clearly stated in the discussion that we cannot account for the possible interference of sperm storage in our study. However, we also note evidence from our captive breeding history that indicates females that are not been bred on an annual cycle frequently lay infertile eggs which suggests sperm storage is not common in the captive setting for this species. Furthermore, we added a sentence to the methods section (Line 170) to make it clear that the complete breeding history of females is not known. We believe we have been sufficiently transparent about this issue in the manuscript and feel this study will still make a valuable contribution to the scientific literature.

Lines 357 to 363 (revised): Sperm storage does occur in snakes, and we cannot discount the possibility of its occurrence here but, based on our experience with captive breeding and artificial insemination experiments (M.R.S. unpub. data), sperm storage is rare in this species. For example, in each of seven years of records from three Louisiana pinesnake breeding facilities, females that were not paired with a male during a breeding season laid unfertilized eggs that year, even if mated in previous seasons, whereas there has been no instance of a non-bred female laying fertilized eggs.

- Please provide better resolution images for Figure 1.

We have adjusted the resolution of the image and hope that the new resolution is adequate for viewing the morphology of sperm as intended.

Discussions and conclusions are adequate

Thank you.

6. PLOS authors have the option to publish the peer review history of their article (what does this mean?). If published, this will include your full peer review and any attached files.

Do you want your identity to be public for this peer review? For information about this choice, including consent withdrawal, please see our Privacy Policy.

Reviewer #1: Yes: Alexandre Rodrigues Silva

Thank you for taking the time and effort to review our manuscript. We appreciate your comments and have identified you in the acknowledgements of the revised manuscript. If you prefer to remain anonymous, we can instruct the journal to remove your name.

---

## [Decision Letter · Decision Letter 1]

24 Feb 2023

Sperm morphology and forward motility are indicators of reproductive success and are not age- or condition-dependent in a captive breeding population of endangered snake.

PONE-D-22-29909R1

Dear Dr. Sandfoss,

We’re pleased to inform you that your manuscript has been judged scientifically suitable for publication and will be formally accepted for publication once it meets all outstanding technical requirements.

Kind regards,

Carlos E. Ambrósio, Ph.D

Academic Editor

PLOS ONE

Additional Editor Comments (optional):

Reviewers' comments:

Reviewer's Responses to Questions

**Comments to the Author**

1. If the authors have adequately addressed your comments raised in a previous round of review and you feel that this manuscript is now acceptable for publication, you may indicate that here to bypass the “Comments to the Author” section, enter your conflict of interest statement in the “Confidential to Editor” section, and submit your "Accept" recommendation.

Reviewer #1: All comments have been addressed

2. Is the manuscript technically sound, and do the data support the conclusions?

Reviewer #1: Yes

3. Has the statistical analysis been performed appropriately and rigorously? 

Reviewer #1: Yes

4. Have the authors made all data underlying the findings in their manuscript fully available?

Reviewer #1: Yes

5. Is the manuscript presented in an intelligible fashion and written in standard English?

Reviewer #1: Yes

6. Review Comments to the Author

Reviewer #1: The authors responded to all of this reviewer's suggestions. In fact, this new version of the manuscript is much more complete, as they provided answers to all the gaps found previously. Thus, acceptance for publication in this form is suggested, unless better judgement.

7. PLOS authors have the option to publish the peer review history of their article (what does this mean?). If published, this will include your full peer review and any attached files.

Reviewer #1: **Yes: **Alexandre Rodrigues Silva

---

## [Editor Report · Acceptance letter]

2 Mar 2023

PONE-D-22-29909R1 

Sperm morphology and forward motility are indicators of reproductive success and are not age- or condition-dependent in a captive breeding population of endangered snake. 

Dear Dr. Sandfoss:

I'm pleased to inform you that your manuscript has been deemed suitable for publication in PLOS ONE. Congratulations! Your manuscript is now with our production department. 

Kind regards, 

on behalf of

Dr. Carlos E. Ambrósio 

Academic Editor

PLOS ONE